# β-Nucleated Polypropylene: Preparation, Nucleating Efficiency, Composite, and Future Prospects

**DOI:** 10.3390/polym15143107

**Published:** 2023-07-21

**Authors:** Bo Wu, Xian Zheng, Wenjie Xu, Yanwei Ren, Haiqiang Leng, Linzhi Liang, De Zheng, Jun Chen, Huanfeng Jiang

**Affiliations:** 1School of Chemistry and Chemical Engineering, South China University of Technology, 381 Wushan Road, Tianhe District, Guangzhou 510640, China; 6066233@163.com (B.W.); dongxiaojun_love@163.com (W.X.); renyw@scut.edu.cn (Y.R.); 2Guangdong Winner New Materials Technology Co., Ltd., Gaoming District, Foshan 528521, China; zxflying1218@163.com (X.Z.); lenghaiqiang@163.com (H.L.); liang-linzhi@163.com (L.L.); gmwinner@vip.163.com (D.Z.); chen_jun_cj@163.com (J.C.); 3The State Key Laboratory of Pulp and Paper Engineering, South China University of Technology, 381 Wushan Road, Tianhe District, Guangzhou 510640, China

**Keywords:** polypropylene, β-crystal, β-nucleating agent, nucleating efficiency, synergism, composite, future prospects

## Abstract

The β-crystals of polypropylene have a metastable crystal form. The formation of β-crystals can improve the toughness and heat resistance of a material. The introduction of a β-nucleating agent, over many other methods, is undoubtedly the most reliable method through which to obtain β-PP. Furthermore, in this study, certain newly developed β-nucleating agents and their compounds in recent years are listed in detail, including the less-mentioned polymer β-nucleating agents and their nucleation characteristics. In addition, the various influencing factors of β-nucleation efficiency, including the polymer matrix and processing conditions, are analyzed in detail and the corresponding improvement measures are summarized. Finally, the composites and synergistic toughening effects are discussed, and three potential future research directions are speculated upon based on previous research.

## 1. Introduction

Polypropylene (PP) was synthesized by Natta for the first time in 1954, and the Montecatini Company of Italy realized its industrialization in 1957. With the rapid development of the PP industry, it has become the thermoplastic general resin with the largest output and the widest use. The macroscopic properties of the material are derived from its microstructure, and the crystallinity, crystal form, and crystal structure of PP play a key role in its properties. The crystalline structure of PP mainly includes α [1], β [2,3], γ [4], δ [5], and the quasi-hexagonal state [6]. α- and β-crystals have been widely studied when compared with the other crystal forms described. In its usual state, PP exists as an α-crystal and belongs to the monoclinic crystal system. Its molecular chain is left-handed, or right-handed when around the central axis. The spiral conformation is the most stable crystal form of polypropylene; meanwhile, the β-crystal belongs to the thermodynamic quasi-stable and kinetically unfavorable crystal form, which requires special conditions in order to obtain it. Padden and Keith first reported the quasi-hexagonal β-crystal of polypropylene in 1959 [7]. Turner-Jones et al. [8] believed that the β-crystal was a hexagonal structure crystal, which influenced the academic community for more than 20 years. It was not until 1994 that Meille [9] and Lotz [10] independently established that the β-crystal was a rhombohedral crystal structure, and that the unit cell parameters were a = b = 11.01 Å, c = 6.5 Å, α = β = 90°, γ = 60°, and a density of 0.921 g/cm^3^. Due to their unique molecular structure, β-crystals have a great influence on the properties of materials, as well as on final products such as the toughness, elongation at break, heat resistance, etc.

In the modification of the polypropylene crystal form, the increase in the toughness caused by the β-crystal makes it a special focus in both academic and industry fields [11]. Chen et al. [12] investigated the mechanical properties of polypropylene homopolymer (PPH), block copolymer (PPB), and random copolymer (PPR), and found that β-crystals greatly improved the toughness of PPH and PPB—whether it was above or below Tg. In a slightly different way, β-crystals may affect the dispersion effect of rubber, but they will not affect the overall toughness of the material in PPR. According to the analysis of the model in this study, the reasons for the high toughness of β-PP are mainly determined by the following points: molecular weight; the density of molecules that are connected between β-crystals (which are especially stable, and tend to form a hollow grid structure when stressed); the arrangement of layers (which can effectively transfer and release stress); and the β–α phase transition (where it is believed that the additional energy absorption mainly comes from recrystallization and a local hardening of the microporous network structure during phase transition) [12].

In addition, the existence of β-crystals allows β-PP to have better mechanical properties, especially in terms of the impact toughness and a better heat resistance than pure PP [13,14]. It was found that the aging time of PP-Rβ, which has a better fatigue crack expansion resistance, was 30% higher than that of PP-Rα [15]. Moreover, the effect of β-crystals on the friction coefficient [16] is less than that of α-crystals.

The present review aims to summarize several aspects, including the influence of β-crystals on the basic properties of materials (especially the toughness of polypropylene); the many methods through which to obtain β-crystals; certain new β-nucleating agents and compounds found in recent years; the influencing factors and improvement methods for β-nucleating efficiency; the application of β-crystal polypropylene composites; and certain potential research directions of β-PP that are detailed on the basis of the previous summary [17,18].

## 2. Preparation Strategies of β-PP

The β-crystal is a thermodynamically unstable crystal form when compared with the α-crystal. It is difficult to produce and to maintain a large number of stable crystal forms under general processing conditions; instead, several special conditions are required.

### 2.1. Common Methods

The common methods that have been used up to the present, through which to obtain β-crystals, are as follows:Temperature gradient—This method not only affects the shape of spherulites but also affects their internal structure. Although it has no effect on the nucleation of spherulite, it can accelerate the transformation from a melting state to spherulite. The density of the crystal nucleus is higher, and it is easier to form β-crystals in regions with larger temperature gradients [19,20].Shear induction—The most intuitive reason for this method’s existence is to cause different degrees of orientation in the polymer molecular chains, thereby obtaining polymers with different properties. Shearing can also induce the regular arrangement of molecular chains, shorten the nucleation time, increase the nucleation density, and induce oriented crystallization in the absence of nucleating agents [21].Quenching and annealing—It has been found, according to previous research, that the formation of β-crystals is evidently dependent on the quenching temperature [7,22]. A certain temperature range is beneficial for the growth of β-crystals [23]. The growth of β-crystals is inhibited, and the growth rate decreases significantly or increases to an α-transformation beyond the temperature range. Ma et al. [24] found that the comprehensive properties of a composite increased with an increase in the annealing temperature, and partial β-crystals were formed synchronously. The annealing process causes a change in the microstructure and greatly improves the impact strength. At a moderate temperature, it can induce the transformation of an α-crystal to a β-crystal.UV light—Zhao et al. [25] found that there was an evident presence of β-crystals after UV irradiation at 65 °C for 600 h, no matter if a pure PP or a composite with zinc oxide were added. Indeed, it was finally proved that UV light is the real reason for the formation of β-crystals.

### 2.2. Addition of β-Nucleating Agent

Adding a β-nucleating agent is considered to be the most effective method through which to obtain β-crystals when compared with the above methods. The γ-crystal linear trans quinacridone (E3B), which was discovered by Leugering [26] in 1967, can induce the formation of β-crystals that have a nearly hexagonal crystal structure in PP at a particularly low content. Unfortunately, PP is dyed red with the addition of E3B (which was the first effective β-nucleating agent). There are many kinds of β-nucleating agents, including inorganic substances [27,28,29,30,31,32], polycyclic aromatic hydrocarbons [26,33], organic acids, salts [34,35,36], amides [37,38,39], and rare earths [40]; however, these agents have already been summarized in previous studies [17,18].

*K_β_* is usually used to characterize the nucleation efficiency and the maximum β-crystal content (*K_β_*), and the index is calculated by the wide-angle X-ray diffraction (WAXD) spectrum. According to the research of Turner Jones et al. [41], the calculation formula is as follows: (1)$$k_{\beta }=\frac{A_{\beta }(300)}{A_{\beta }\left(300\right)+A_{\alpha }\left(110\right)+A_{\alpha }\left(040\right)+A_{\alpha }(130)}$$

*A_α_* (110), *A_α_* (040), and *A_α_* (130), respectively, represent the area of the diffraction peak of the (110), (40), and (130) characteristic peaks of the α-crystal on the WAXD spectrum when it corresponds to 2θ, and these are 14.1°, 16.9°, and 18.6°, respectively. *A_β_* (300) corresponds to the only (300) diffraction peak of the β-crystal, where 2θ is 16.1° (as shown in Figure 1). It was disclosed that all the four samples exhibited two characteristic diffraction peaks at 2θ =16.1° and 21.2°, which correspond to the (300) and (301) crystal planes of the β-form. Furthermore, the diffraction peaks at 14.1°, 16.9°, and 18.6° (which correspond to α (110), α (040), and α (130)) were undetectable, indicating that all the four precursor films were composed of almost pure β-crystals, as shown in Figure 1 [42].

Of course, the content of β-crystals can also be calculated by a differential scanning calorimetry (DSC) curve, but there are shortcomings to this calculation formula, which were described in the study of [18]. In addition, it was found that the total crystallinity value of the sample that contained a nucleating agent when measured by WAXD is higher than that of the same sample measured by DSC. *K_β_* is usually calculated by researchers according to the WAXD data. This is due to the fact that the crystal is thermally unstable and that the “decrystallization” [43] occurs during the heating process of DSC. 

#### 2.2.1. New β-Nucleating Agents or Compounds for Polypropylene

A variety of new β-nucleating agents or compounds with a high β-nucleation effect are listed in the table below. These agents have been found by researchers in recent years through the activity and development of research. Different nucleating agents have different efficiencies in inducing polypropylene to form β-crystals. Of course, the results of these procedures are related to the type, even the brand, of polypropylene, as well as the addition amount of the nucleating agent. Therefore, these differences are also explained in the following table.

According to Table 1, the simplest type of β-nucleating agent is a kind of heat-treated shell, and its main component is calcium carbonate. The addition of 5% of this β-nucleating agent can obtain an 80.1% β-crystal conversion rate, and the mechanical properties are also greatly improved, according to its description in the literature. This is a particularly cheap and easy-to-obtain β-PP modification case. In addition, the industrialized nano-zinc oxide, zinc tetrahydrate, and DCHT from Shanxi [44] can obtain β-crystals with a conversion rate of more than 95%, which is an especially important choice for β modification.

As a simple and easy processing method, adding a nucleating agent is the most direct and effective means through which to obtain β-PP. The induction mechanisms of β-nucleating agents mostly conform to a heterogeneous nucleation mechanism. The dispersion of a β-nucleating agent [64] in the matrix and orientation structure (or ordered structure) of a polymer melt [65] are two key factors that affect the formation of β-crystals. The former affects nucleation sites, while the latter determines β-crystal growth. According to the research of Binsbergn [66,67,68], the nonpolar part of the nucleating agent forms dents on the surface, and this can accommodate the embedding of polypropylene chain segments, as well as arranging them in an orderly fashion. The nucleating agent acts as a crystal nucleus that reduces the growth time of the crystal nucleus itself and accelerates the crystallization rate significantly. A large number of heterogeneous crystal nuclei will induce a large number of spherulites at the same time. A large number of spherulites will also inevitably come into contact with other spherulites and stop them growing in the process of their perfect growth, which results in most sizes of spherulites being smaller than those obtained by the homogeneous nucleation of PP itself. Therefore, polypropylene with a large number of spherulites that are of small size and high crystallinity is finally obtained by adding a nucleating agent, and its performance is also greatly improved. The crystallization process is shown in Figure 2.

In addition, the morphology of nucleating agents also plays an important role in crystallization [44]. The researchers in a previous study [42] used four typical β-nucleating agents; as such, four β-crystal morphologies were, respectively, obtained, as shown in Figure 3 below. The β-isotactic polypropylene (iPP) precursor film is composed of “bundle-like” lamellae without fully developed spherulites when in the presence of NAB83 (as shown in Figure 3a). The β-hedrites were formed by adding Pa-Ca (shown in Figure 3b), and the “flower-like” β-crystals were detected in the presence of WBG-II (shown in Figure 3c). On the other hand, the β-iPP precursor film containing TMB-5 (shown in Figure 3d) was composed of fully developed β-spherulites [42].

On the other hand, a β-nucleating agent was chemically loaded onto the surface of inorganic materials such as calcium carbonate, multilayer carbon nanotubes, and graphene. The final composite not only has a high β-crystal content, but also excellent properties, which give full play to the synergistic effect between substances. The main content of this will be mentioned in the following chapters.

#### 2.2.2. Polymer β-Nucleating Agent for Polypropylene

Most of the β-nucleating agents used for research and commercialization are small organic molecules; however, they have certain disadvantages at present. For example, they are easy to agglomerate and have poor dispersibility during polymer processing. Certain nucleating agents also need surface treatment or the addition of dispersants in order to obtain a better dispersion effect, while others are easier to precipitate. Polymer β-nucleating agents have gradually become a research hotspot in recent years. The following Table 2 lists some of the current polymer β-nucleating agents, as well as listing the type and brand of polypropylene; furthermore, the additional amount of nucleating agent is the same as the above.

The β-nucleation induction efficiency of polymers is generally low, and the addition amount is usually more than 1%. At present, it is known that more than 90% of β-crystal conversions produce a kind of liquid crystal with a special structure, but there is currently no industrial product that can be compared with small-molecule organic or inorganic β-nucleating agents. There is still more research work to be carried out.

The research of polymer β-nucleating agents is just starting to expand. It is of great theoretical significance to systematically study the nucleation efficiency and nucleation mechanism of polymer nucleating agents. At the same time, further investigation of the two-phase or multiphase structure after adding polymer nucleating agents will actively promote the development of new PP alloys with high performances.

Compared with the common methods, adding β-nucleating agents to obtain βPP is indeed simpler and easier to implement. Many of the small-molecule and polymer β-nucleating agents reported in the literature are listed. Among these β-nucleating agents, β-PP can be obtained by selecting the appropriate nucleating agent according to the reported β-crystal conversion rate. Of course, industrially mature varieties such as nano-zinc oxide, zinc tetrahydrate, and DCHT from Shanxi may be a better choice. On the other hand, for the study of PP alloys, polymer β-nucleating agents may provide more research ideas and directions.

## 3. Nucleating Efficiency of β-Crystals

### 3.1. Influencing Factors of β-Nucleating Efficiency

Different nucleating agents have different β-nucleation efficiencies in polypropylene, even with the same addition amounts. On the other hand, a nucleating agent has different β-nucleation efficiencies in different kinds of polypropylene with the same addition amounts. This is related to the fact that the β-nucleation induction efficiency is affected by many factors.

#### 3.1.1. Chain Structure

It is a fact that chain irregularities are known to negatively affect β-phase formation. Wang et al. [11] used several different isotactic polypropylene homo- and copolymers based on the Ziegler–Natta and single-site catalysts, which differ mostly in chain-defect concentration, to evaluate the β-nucleating agent “WBG” sensitivity’s chain structure effects. They found that there was a higher sensitivity toward the chain regularity of isotactic polypropylene, as well as a more limited impact strength increase.

In addition, long-chain branching also leads to the formation of a polycrystalline state, which induces the formation of γ-crystals and inhibits the formation of β-crystals. When a 5% long-chain-branched polypropylene (LCB-PP) is added to the system, the formation of β-crystals is greatly inhibited. LCB-PP increases the crystal density through the self-cleaning effect. Therefore, there are a large number of mixtures of the α and γ phase in the system. They have a higher thermodynamic stability than β-crystals and are more easily induced in systems with long-branched chains [79]. On the contrary, in a high β-crystal-content system, the β-crystals, nucleating agents, and LCB induce particularly small spherulite sizes. The introduction of a nucleating agent and a long-branched chain improves the impact toughness without reducing the yield strength and elongation at the break. When both exist at the same time, there is a synergistic toughening effect [80].

The processing effect and the change in morphology will cause a difference in the β-crystal content [81,82,83]. The chain structure changes the number of crystal centers and affects the spherulite size. The irregularity of the chain segments in the stereoscopic and regional effects as well as the insertion of comonomers have a negative effect on the β-crystal efficiency in the polymer matrix.

#### 3.1.2. Ethylene Phase

Isotactic polypropylene includes the presence of homopolymers (PPH), block copolymers (PPB), and random copolymers (PPR), as described in the study of [84]. Melt recrystallization seems to occur only in PPs that stretch with little change in their crystallinity in copolypropylene (coPP). The final molecular orientation of the stretched film showed a significant linear downward trend, and the ethylene segment causes most of the copolymerized polypropylene to form many γ-crystals, which are transformed into α-crystals after stretching [85]. It was found that the β-nucleation efficiency of both block copolymerized PPBs and homopolymerized PPHs was significantly higher than that of random copolymerized PPRs. The half-crystal time of block copolymerized PPs was much lower than that of random copolymerized PPRs. Some of the ethylene segments in a random copolymerization reduced the crystallization rate [86,87]. The relative fraction of β-crystals decreased with the increase in the ethylene comonomer content [88]. The reason for this is not only related to lattice matching [89,90], but also to the important relationship that it has with morphological characteristics, which depends on the molecular interaction at the crystallization interface [91].

Studies in recent years have found that the ethylene phase has a priority “selectivity”. Peroxides preferentially attack the monotertiary hydrogen that is adjacent to one or more ethylene units, those that are between ethylene units, or those at the end of polypropylene blocks, resulting in the selective functionalization of ethylene-rich copolymers (which is independent of the solubility parameters or decomposition rate of peroxides). Degradation and functionalization mainly occur in the ethylene-rich phase [92]. On the contrary, the content of β-crystals and the molecular weight decreases at the same time that the impact of the mechanical properties gradually decreases with the increase in peroxide. The decrease in the fracture strength is related to the decrease in the molecular weight and amorphous-phase-chain entanglement. According to this, the degradation of peroxide is unfavorable when attempting to obtain βPPR [93].

Fan et al. [94] observed different phenomena. Their K_β_ value was up to 0.82 with an increase in the dicumyl peroxide (DCP) content. It was unexpectedly found that the β-nucleation increased, which may be related to the improvement in the stereoregularity of controlled-rheological polypropylene random copolymers (CRPPR) with the increase in DCP. It was found that the degraded PPRs have a thicker lamellar layer, which results in a component with improved stereoregularity. The authors speculate that the possible degradation mechanism has free radicals that are more likely to attack the tertiary carbon atoms that are close to the ethylene comonomer. Furthermore, the ethylene unit is only at the end, and there is basically no ethylene comonomer in the middle, thus improving the stereoregularity of the degraded PPR and ultimately affecting the β-crystals of PPRs in reactive extrusions.

According to the above studies, it was found that the ethylene phase and ethylene chain in the polymer also had an important influence on the formation of polypropylene β-crystals. Whether in copolymers or random copolymers, with the introduction of ethylene segments, the intuitive result is that the β-nucleation sites or crystallization centers of polypropylene are reduced. Therefore, the addition of a β-nucleating agent to polypropylene that contains ethylene segments can usually only obtain a worse β-nucleating crystallization effect than those of homopolymerized polypropylenes.

#### 3.1.3. Polymorphism

Another problem affecting the nucleation effect of β-crystals is the coexistence of different polymorphic forms of polypropylene in addition to the great influence of the ethylene chain segment. It is well known that the α, β, γ, and other five crystal forms of polypropylene, which are unstable and prone to crystal transformation, need certain conditions (except for the conventional α-crystal) in order to be obtained. However, the coexistence of different polymorphic forms will still affect the main crystal form, just as it is difficult to obtain a 100% β-crystal PP [43,82]. It is possible to obtain γ-crystals [95,96] under different crystallization conditions in both polymers and composites. Certain nucleating agents, including β-nucleating agents, with a strong crystallization induction ability also have a dual nucleation ability [11,97,98]. The increase in the α and γ-crystal content leads to a decrease in the β-crystal content [99,100,101] when the total crystallinity is fixed. Fu et al. [102] found that if the defects of the copolypropylene molecular chain cannot be overcome by a large number of β-nucleating sites, then γ-crystals can be formed.

Based on the facts detailed above, the influence factors of α- or γ-crystals should be minimized. Increasing the nucleation site of β-crystals or appropriate processing conditions can make the induction effect of β-crystals dominant. This helps to obtain polypropylenes with a high β-crystal content, despite other polymorphs of PP also existing.

#### 3.1.4. Interface Action

There are different β morphologies with different nucleating agents, according to the study of Yang et al. [41] The polymer crystallizes epitaxially on the surface of the nucleating agent, which affects the folding and arrangement of the polypropylene segments, thus forming different crystal forms [103].

Köpplmayr et al. [104] studied the thermomechanical properties of βPP-multilayer films, which were prepared by the uniform alternation of 128 layers of PPR and βPPR. It was found that the single-layer interface showed a nucleation effect, and the content of β-crystals increased with an increase in the number of layers. According to the temperature dependence of the flexural modulus, the number of high rises will lead to an increase in the stiffness in a large temperature range. PP and PE are incompatible systems [105]. Polymorphic PP is the infiltration phase, and PE is the infiltration layer when using them in melt infiltration engineering. The competition between the surface-induced effect and the shear-induced crystallization during the melt infiltration process produces a phase morphology transition from string crystal to columnar crystal [106].

When the semicrystalline polymer is confined to isolated micro or nano domains in the mixture, nucleation can control the whole crystallization dynamically in a sufficiently small region; this has a positive contribution, especially for immiscible blends [107,108].

#### 3.1.5. Processing Conditions

The processing of polymers is usually accompanied by processing conditions such as the shear force, temperature, and pressure. The β-crystal is a thermodynamically unstable form, and the processing conditions have a great influence on the formation of β-crystals in the matrix. Inappropriate processing conditions will still inhibit the formation of β-crystals, even if there is a highly efficient β-nucleating agent.

As mentioned above, it has been known that shear can induce polypropylene to produce different crystalline states under different conditions [109]. Generally speaking, β-crystals will gradually change into α-crystals, γ-crystals, or mesophases until they finally disappear with the increase in the shear rate or processing rate [110,111,112]. Shear can produce β-cylindrical crystals, called β-cylindrutes, in the process of controllable rheological polypropylene processing [113].

When stretching the film, voids [114,115] will be generated during biaxial stretching, and the phase shape and crystal form will also change. In addition, obvious melt recrystallization may occur during the stretching process [116]. However, the tensile process of an injection-molded sample seems to produce different changes [117]. The surface shear region is mainly oriented on α-crystals and partial γ-crystals, and the middle region is enriched with β-crystals. In general, tensile deformation will promote the change in β-crystals to α-crystals, and the change in volume will produce a uniform distribution of pores. The porosity increases with the increase in the β-crystal content.

A continuous and orderly alternating of the α–β-crystal PP layer is formed by temperature control in a multilayer PP sheet. A large number of β-crystal layers can be obtained by isothermal crystallization at 130 °C and 50 °C/min, and the content of β-crystals is 24.6% when this is performed. However, only α-crystals are formed when the cooling rate is changed to 1 °C/min [118]. In the heat treatment process, β–α-transformation occurs. The heat treatment temperature of oriented β-crystals is divided into three stages. The increase in β-crystals comes from a secondary crystallization below the melting temperature of the oriented β-crystal. When the temperature is between an α orientation and β orientation, there is mainly an α orientation. When the temperature exceeds the α orientation, β-crystals are produced again [119].

Pressure plays the role of a generalist in the presence of shear flow as shown in Figure 4 [120]. A large number of β-crystals can be obtained in the range of a low pressure of 5 MPa and a shear rate 0–24 S^−1^. Low shear can significantly inhibit the production of β-crystals when the pressure and shear rate are 50–100 MPa and 3.2 S^−1^. There is no β-crystal that can be produced when the pressure is 150 MPa. The processing window of pressure and shear flow is summarized for the first time, and this provides an important reference value for the processing conditions of βPP. In addition, pressure also induces the formation of different crystalline states. When the pressure increases, the β- and γ-crystals decrease and the β-nucleating agent can also induce the formation of γ-crystals in a wide pressure range [121].

This study also found that microwave irradiation can lead to the occurrence of β–α-transformation [122]. Under microwave irradiation, the total crystallinity decreased by 5–6% with a decrease in the β phase. Parts of the β-crystals transformed into α-crystals, and other parts were transformed into an amorphous state. The β phase decreased and the impact gradually decreased due to the increase in the amorphous region.

### 3.2. Improvement in β-Nucleating Efficiency

The nucleation efficiency of β-crystals is affected by many factors. It is not an easy thing to obtain a PP with a high content of β-crystals. In addition, during induction growth and processing, β-crystals may be transformed into α-, γ-, or even mesophase crystals [123,124,125] under different shear force and thermal conditions. How one is to improve the nucleation efficiency of β-crystals and obtain a high content of β-crystal polypropylene has always been the direction of people’s efforts.

#### 3.2.1. Processing Conditions

Many things have two sides. Shear, temperature, and pressure inhibit the formation of β-crystals as described earlier. Interestingly, they can also produce synergistic effects and can promote β-crystal growth, including in a β-nucleating agent system that usually has an optimal crystallization nucleation condition [126].

It was found that the synergistic effect of annealing and nucleating agents improved the toughness of PPB [127]. The molecular chain dynamic capacity of the amorphous region in the matrix determined the toughness of the material. Whether or not there is rigidity in the material of α-crystals or the toughness of ductile β-crystals can be improved by changing the distribution of different phases. The synergistic effect of nucleating agents and annealing is more significant when seeking to improve this effect. Annealing promotes the cavitation of the amorphous β-phase, and the subsequent shear of the crystalline phase in PP is mainly achieved by increasing the number of rigid amorphous fractions (RAFs) and decreasing the density of mobile amorphous fractions (MAFs). Therefore, this will cause large-scale plastic deformation. In addition, the perfection of the crystal also contributes to the cavitation before crystal shearing. These changes that are caused by annealing lead to a final change in the β-crystal structure and an improvement in the material properties [128]. Of course, a proper addition and a sufficiently low cooling rate also help β-crystal formation [129].

Solid-state tensile deformation induces crystallization. When the tensile temperature is higher than the crystallization temperature, the β-lamellae are fractured and the columnar structure is formed. With the increase in the temperature, β–α transition occurs based on the existence of stress relaxation. The phase transition process of the crystal is beneficial for improving the mechanical properties of the material, and solid-state stretching can improve the toughness and strength [130].

#### 3.2.2. Promotion of Homopolymers

Isotactic homopolymers usually obtain a higher β-crystal content than copolymers and random copolymers, as mentioned above. Further studies have found that the crystallization ability is also affected by the molecular weight and isotacticity. When the molecular weight is above a certain degree, the higher the isotacticity and the wider the molecular weight distribution, the stronger the β-crystallization ability is [131,132]. Therefore, the introduction of homopolymers in copolymers must be a good choice.

Menyhárd et al. [133] studied the blend systems of iPP/sPP, iPP/rPP, iPP/PVDF, and iPP/PA-6. They determined that the key factor for the formation of β-crystals is the α nucleation efficiency of the second polymer. When the polymer has an α nucleation, the crystallization temperature range is lower than that of β-PP. Based on this, the author draws an important conclusion that has also affected many subsequent researchers, that is, the content of β-crystals in the copolymer can be increased by adding homopolymerized PP.

The introduction of PPR into isotactic polypropylene or impact polypropylene compound nucleating agents increases the impact strength of the system without reducing the rigidity [81,134,135]. After the addition of isotactic PP, the site of β-crystals is increased, and the homopolymer PP first becomes a β-nucleating center. The crystallization is then induced by the β-nucleating agent so that the rigidity can be maintained, and the toughness can be significantly improved.

#### 3.2.3. Modification of Nucleating Agent

The improvement in the nucleating agent includes various means such as loading to improve dispersion, catalystization, dynamic in situ reaction, and activation treatment. The main purpose is to improve the dispersion of nucleating agents in the polypropylene matrix by using various effective methods through which to obtain more crystallization sites and to further improve the nucleation efficiency of β-crystals.

The dispersion of nucleating agents in a PP matrix is improved, and a synergistic effect is produced by loading nucleating agents on different carriers [136,137]. For example, pimelic acid reacts with calcium carbonate to form calcium pimelate, and this is then loaded with nano calcium carbonate. The impact resistance of the system is greatly improved because of the high content of β-crystals. The impact fracture surface appears, interestingly, to be filamentary, although the impact fracture surface containing calcium pimelate and complex is smooth [138].

The combination of nucleating agents is also an effective method. Zhu et al. [139] used adipic acid to treat calcium carbonate (AA-RCC) and combined it with the rare earth β-nucleating agent WBG. It was found that linear crystals begin to appear in the initial stage of nucleation. The α-crystal appeared before the β-crystal. It grew outward symmetrically to form a network structure with uniform density and a close arrangement with the passage of time. Specially arranged α-crystals induced by AA-RCC as nuclei can induce the formation of β columnar crystals. The α-crystal will gradually transform into a β-crystal when the spherical crystal grows faster than the columnar crystal. The content of β-crystals was basically the same as those in the slow crystallization process when subjected to rapid cooling, and this process is achieved by adding a 0.02% α-nucleating agent and a 0.2% β-nucleating agent in the field of industrial extruded pipes. The comprehensive performance of the pipe was improved, which provides a new direction for pipe extrusion [140].

A bifunctional catalyst for the preparation of β-polypropylene was constructed by introducing a β-nucleating agent of polypropylene into the Ziegler–Natta catalyst. The catalyst had good stereospecificity for polypropylene, which can make the isotacticity of the polymer reach 97–99%. The catalyst is sensitive to hydrogen and has a strong molecular weight adjustability. The “fragmentation–refinement–dispersion” of the nucleating agent can be achieved during the polymerization process through the “catalyst–polymer morphology replication effect”. A β-crystal-modified PP with a nano size dispersion and a more stable scale was obtained [141].

The main nucleating agent in a dynamic in situ reaction is carboxylate or fatty acid salt [142,143,144,145,146,147]. One of the raw materials is liquid, and the reaction temperature is not high (i.e., where the melting temperature of polypropylene can be satisfied). Liquid carboxylic acids or fatty acids can be fully contacted with metal oxides during the dynamic melting process. The generated nucleating agent is further dispersed in the polypropylene melt through the shearing action of the twin screw, and this is performed in order to obtain a βPP with good dispersion and an outstanding nucleation effect. This method also provides a reference for other nucleating agents through which to solve the problem of agglomeration and dispersion. 

The surface activity treatment [148] of nucleating agents can also further improve the nucleation efficiency, dispersion, and self-assembly aggregation [149] of β-nucleating agents. A surfactant connects PP and nucleating agents to improve compatibility. In addition, it was found that ultrasonic vibration can also improve dispersion and can improve the efficiency of rare earth β-nucleation [150]. When the ultrasonic distance is 1 cm, the best dispersion and the highest nucleation efficiency are obtained.

#### 3.2.4. Synergetic Effect

Researchers in recent years have invested more research in the β synergistic combination of nucleating agents and other materials. The combination of β-nucleating agents and suitable materials can often obtain a synergistic effect, as well as further improve the nucleation effect of the β-nucleating agents.

Synergistic materials with β-nucleating agents include MWNTs [151]; illite [56]; calcium sulfate whiskers [152]; nano-silica particles [153,154]; the PA66 used in sandwich-assembly interlayer PP [155]; TPE/calcium carbonate [156]; the polyhedral oligomeric silesquioxane (POSS) amide group hydrogen bond reaction [157]; hydrotalcite (HT) [158]; and even new nanomaterials such as Mxene [159]. β-nucleating agents can maintain better performances in the system via their synergistic effect with different materials. Inducing β-crystals and increasing β-crystal content are the main reasons for the increase in the toughness.

In summary, in addition to the nucleation efficiency of the nucleating agent itself, the chain structure of polypropylene, the ethylene phase, processing conditions, and other factors will affect the β-crystal conversion efficiency. Correspondingly, the nucleation efficiency of the nucleating agent itself can be improved by modification or through synergistic compounding. Furthermore, the introduction of homopolymers can also increase the crystallization sites and reduce the influence of the ethylene phase. At the same time, the selection of appropriate processing conditions will help to obtain polypropylene with higher β-crystal conversions.

## 4. Composite of PP with β-Nucleating Agents

At present, the application of polymers is mostly based on composite materials. The introduction of a variety of inorganic fillers in the preparation process can overcome certain shortcomings and can obtain composite materials with better comprehensive performances. This field has achieved good development in recent years.

Adding β-nucleating agents and impact polypropylene copolymers (IPCs) at the same time can induce a synergistic toughening effect, although carbon nanotubes do have the effect of inhibiting β-crystals [151,160]. Interestingly, in the PPR/SBS system, multilayer carbon nanotubes act as a bridge between PP and SBS in order to absorb the impact energy [161].

Graphene usually has a reinforcing effect on polypropylene [162]. Graphene nanosheets modified by pimelic acid are used as effective β-nucleating agents. Chemical modification increases the compatibility with polypropylene. The composite obtained a higher storage modulus, and the impact strength increased by about one. Hyperbranched polyester-grafted graphene oxide (GO) increased the content of the α phase and the average crystal particle size, thus increasing the crystallinity of the system [163].

The toughness, elongation at break, and heat resistance of the wood-flour-filled polypropylene composites were improved by introducing β-nucleating agents [164,165,166,167,168]; this also indicates the direction for the application of wood-flour-filled polypropylene composites.

Long-glass fiber-reinforced material is an important category of plastic in comparison to steel. The impact strengths of β-PPR and β-LGF/PPR/MPPR are increased by 50.1% from 13.92 to 20.89 KJ/m^2^ after the introduction of a β-nucleating agent, which further broadens the comprehensive performance of composite materials and expands their industrial application range [169].

### 4.1. Toughening Mechanism of Composites

It has been concluded, according to certain studies, that the increase in the toughness is mainly due to the contribution of β-crystals. The loose lattice structure makes the rubber phase more dispersed, and it can further accommodate the hard segments of the toughened rubber phase. The increase in the β-crystal content and the thickening of lamellae also contribute to energy absorption [170,171,172]. On the other hand, β-grain refinement reduces the stress concentration in the crystal region, while the rubber phase improves the impact toughness. The homopolymerization sequence is easier to insert into isotactic PP when seeking to consolidate this structure [173]. The presence of nucleating agents induces the formation of different supramolecular structures. More perfect β-crystal structures can improve the impact toughness, while the partial aggregation of β-crystals can also compensate for the loss of strength [174]. In addition, inorganic nanoparticles act as stress concentrators and establish stress fields around them. The stress field makes the adhesion between the particles and the polymer matrix weak; thus, debonding occurs at the particle–matrix interface. This leads to the release of strain constraints at the crack tip, which leads to a large amount of plastic deformation and consumes a great deal of energy [175].

On the other hand, certain studies also believe that the toughness mainly comes from the dispersion of the rubber phase and the improvement in the chain segment movement ability. The particle size of rubber particles decreases after adding β-nucleating agents into PPR. No matter whether the matrix crystal is an α or β-crystal, it has no main contribution to the impact toughness. The actual contribution comes from the movement of the chain segments in the amorphous region of polypropylene. The addition of β-nucleating agents will increase the movement of the chain segment, which is temporarily attributed to the combined effect of heterogeneous composition and morphology [176,177]. In addition, the transformation from α-crystals to β-crystals reduces the plasticity of the PPR matrix, which makes the matrix more amenable to shear yield during the impact process. Rubber with a higher molecular activity and good dispersion at low temperatures contributes positively to the shear yield of the PPR matrix, which greatly improves the impact toughness [178,179]. It is believed that the synergistic effect of a high β-crystal content and the good dispersion of the rubber phase improves the toughness of the system [180,181,182,183]. The fuzzy-phase interface and craze-shear band of β-crystals can absorb a great deal of energy. The 3D-printed honeycomb materials also confirm this view [184]. The compound with the highest β-crystal count has the strongest impact. High-impact composites can be prepared via the synergistic effect because the addition of a nucleating agent reduces the addition of filler.

### 4.2. Post-Consumer Polypropylene with β-Nucleating Agents

Post-consumer recycled (PCR) materials are “extremely valuable”. The waste plastics generated after circulation and consumption can be turned into valuable industrial raw materials through physical or chemical recovery when seeking to realize resource recycling. In view of the relationship between the content of β-nucleating agents, the relative content of the β-phase, and the recovery cost, the addition of β-nucleating agents is finally controlled below 0.7% in the high-value application of post-consumer polypropylene (PCW-PP). The introduction of β-nucleating agents effectively improves the comprehensive performance of PCW-PP [185]. This method has great application potential in the high-value-added recovery of PCW-PP. The recovery value of wollastonite-filled PP was improved by adding β-nucleating agents directly or by adding wollastonite that was loaded with a β-nucleating agent in order to obtain a higher β-crystal content, good comprehensive mechanical properties, and low-cost consumption [186]. The above work has important social and ecological significance.

By analyzing the toughening mechanism of the composite with β-nucleating agents, it was shown that β-nucleating agents have almost no negative effects when they are used together with other fillers such as IPC and glass fiber, etc. On the contrary, they all improve the comprehensive mechanical properties of the material, and they have a positive effect in obtaining better composite materials and PCR materials.

## 5. Future Prospects of β-PP

The following, with the accumulation and development of research, may be the new development direction of β-PP research in the future.

PP has a self-nucleation ability after the process of changing the grafting group to the end group. This phenomenon is different from the traditional nucleating agent system and the long-branched polymer system [187]. If this system is compounded with nucleating agents, will it have a better effect? Of course, the synergistic mechanism of nucleating agents is still an important research direction to be considered in the future.

The research of polymer β-nucleating agents is in the ascendant, and this is not the case for small-molecule β-nucleating agents. The introduction of ionic monomers into polymers helps to further improve the nucleation ability [71], which helps to design more polymer nucleating agents in the future. It is of great theoretical significance to systematically study its nucleation efficiency and nucleation mechanism. At the same time, further investigation of the two-phase or multiphase structure after adding polymer nucleating agents will actively promote the development of new PP alloys with high performances.

The crystal morphology of the cast film was controlled during processing, and it was attempted to find out the crystal morphology suitable for biaxial stretching. The pores and morphological distribution caused by β–α-transformation are still attractive to researchers, even though researchers have made achievements in the field of microporous film research [116,188].

## 6. Conclusions

The β-crystal is a metastable crystal form of polypropylene, and it can bring about changes in various physical properties. The formation of β-crystals can improve the toughness and heat resistance of materials. Compared with the common methods, adding β-nucleating agents to obtain βPP is indeed simpler and easier to implement. Many of the small-molecule and polymer β-nucleating agents reported in the literature are listed in this study. The industrialized nano-zinc oxide, zinc tetrahydrate, and DCHT from Shanxi can obtain β-crystals with a conversion rate of more than 95%. On the other hand, for the study of PP alloys, polymer β-nucleating agents that have not been industrialized may provide more ideas and directions. The chain structure of polypropylene, the ethylene phase, processing conditions, and other factors will affect the β-crystal conversion efficiency. Correspondingly, various methods can improve the conversion rate of β-crystals, such as the introduction of homopolymers, the modification of nucleating agents, the selection of appropriate processing conditions, etc. Subsequently, β-nucleating agents have a positive effect when they are used together with other fillers such as IPC and glass fiber, etc. Inorganic nanoparticles and β-crystals as fillers are helpful in improving the toughness—especially regarding the synergistic effect on PCR materials, which is of great social and ecological significance. Finally, synergistic compounding, polymer nucleating agents, and microporous film are considered three important directions for β-PP in future research.

## Figures and Tables

**Figure 1 polymers-15-03107-f001:**
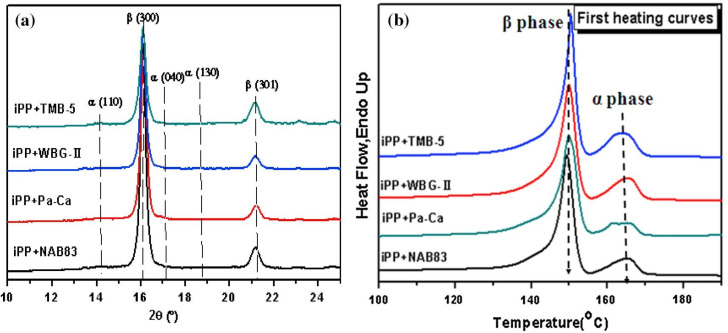
(**a**) WAXD and (**b**) DSC diagrams of the β-iPP cast film, which contain four different nucleating agents [42].

**Figure 2 polymers-15-03107-f002:**
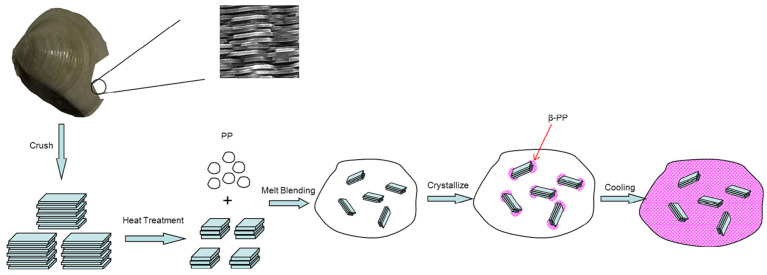
The generation progress of the β-PP induced by the monetaria moneta powder [45].

**Figure 3 polymers-15-03107-f003:**
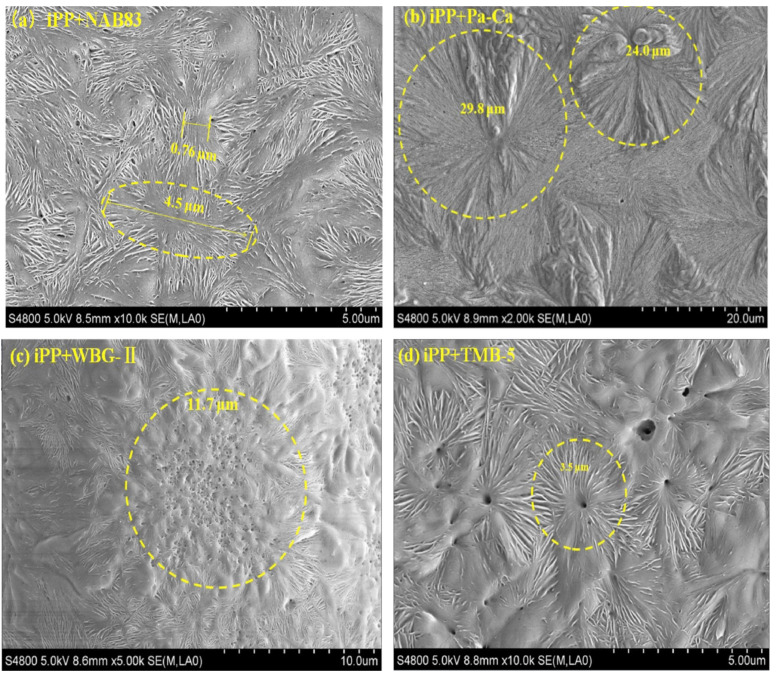
The different morphologies of β-crystals induced by different nucleating agents [42].

**Figure 4 polymers-15-03107-f004:**
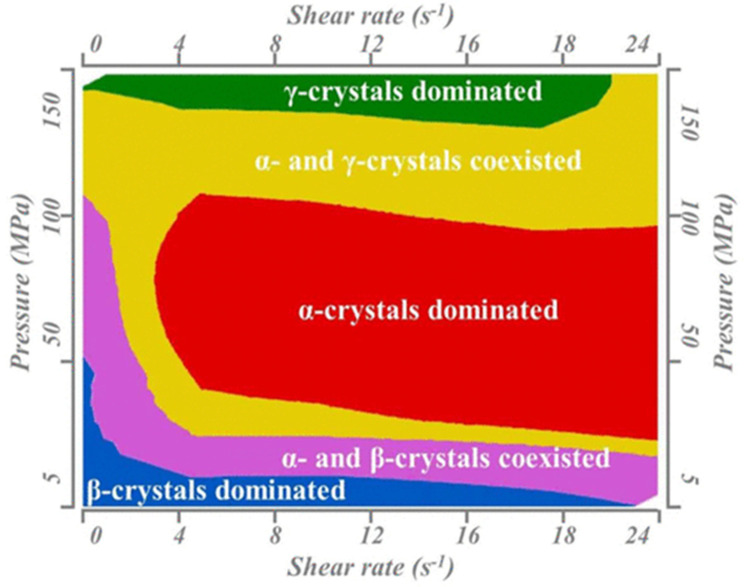
The window of pressure and flow for producing β-crystals in isotactic polypropylene [120]. Reprinted (adapted) with permission from [120]. Copyright {2017} American Chemical Society.

**Table 1 polymers-15-03107-t001:** New β-nucleating agents or compounds.

Sequence	Nucleating Agent	Provider	*K_β_*/%	Polypropylene Category	The Best *K_β_* Amount Added %	Reference
1	Calcium carbonate	Heat-treated shell	80.1	Isotactic polypropylene, grade F401	5	[45]
2	P-cyclohexylamide Carboxybenzene	Prepared in laboratory	96.96	Isotactic polypropylene, grade T30S	0.05	[46]
3	Zinc tetrahydrate	Guangzhou Chenghe technology company (China)	97.2	Isotactic polypropylene, grade T30S	0.1	[47]
4	Established lignin zinc salts	Lignin powder	92.77	Isotactic polypropylene	0.2	[48]
5	Nano-zinc oxide	Jing Rui new material Co., Ltd. (China)	95.2	Isotactic polypropylene, grade T30S	3	[49]
6	*N*,*N*′-dicyclohexylsuberoylamide and *N*,*N*′-dicyclohexylsebacoylamide	Synthesized in laboratory	Not mentioned	Homopolymer, H649 Heterophasic copolymer K 693 Random copolymer R 605	Not mentioned	[50,51]
7	Pimelic acid supported chemically on treated keratin fibers	Industrial waste	79	Isotactic polypropylene	0.5	[52]
8	Zinc suberate	Synthesized in laboratory	82	PPR powder, brand T4401	0.2	[53]
9	Bulk molybdenum disulfide	Composites Innovation Centre (Canada)	Not mentioned	Isotactic polypropylene	Not mentioned	[54]
10	Calcium tetrahydrate	GCH Technology Co., Ltd. (China)	93.5	Impact resistant polypropylene copolymer brand j340	0.03	[55]
11	Cadmium bicyclo[2.2.1]hept-5-ene-2,3-dicarboxylate (BCHE30)	Synthesized according to a patient in laboratory	87	Isotactic polypropylene, grade F401	0.1	[36]
12	*N*,*N*′-dicyclohexylterephthalamide (DCHT)	Shanxi Provincial Institute of chemical industry (China)	0.95~1.0	Isotactic polypropylene, grade S1003	0.05	[44]
13	Illite modified by calcium heptaneate	Chemical modified in laboratory	93.31	Isotactic polypropylene	5	[56]
14	*N*,*N*′-dicyclohexyl-1,5-diamino-2,6-naphthalenedcarboxamide chemically supported on the surface of MWCNTs	Synthesized in laboratory	93	Isotactic polypropylene, grade T30S	0.05	[57]
15	Alkyl-substituted benzoate alumina	Synthesized in laboratory	>80	Homopolymer	0.2	[58]
16	N1, N4-Bis (2,2-dimethylbutyl) terephthalamide (TPA-CP) N1, N4-Dicyclohexylterephthalamide (TPA-CA) N1, N4-Dicyclopentylterephthalamide (TPA-CP) N1, N6-Diphenyladipamide (ADA-PA)	Synthesized in laboratory	Not mentioned	Isotactic polypropylene powder PP-HGD	0.2	[59,60]
17	Hexahydroxythalic barium	Synthesized according to a patient in laboratory	80.2	Isotactic polypropylene, grade T30S	0.4	[61]
18	Silesquioxane functionalized with *N*,*N*′-dicyclohexyl-2,6-naphthalene Dicarboxamide (SF-B01)	Department of organic chemistry UAM (Poland)	84.77	Isotactic polypropylene, grade hp500n	0.25	[62,63]

**Table 2 polymers-15-03107-t002:** New polymer β-nucleating agents.

Sequence	Nucleating Agent	Provider	*K_β_*/%	Polypropylene Category	The Best *K_β_* Amount Added %	Reference
1	Zinc polyacrylate Potassium polyacrylate Sodium polyacrylate	Synthesized in laboratory	12 18 25	Isotactic polypropylene powder PP-HGD	0.3	[69]
2	Poly (acrylonitrile–butadiene–styrene) (ABS)	Chimei Industrial Co., Ltd. (Taipei, Taiwan, China)	36.2	Isotactic polypropylene, grade T30S	2	[70]
3	Liquid crystal polymer ionomer with 5 sulfate monomers (PBDPSi5)	Synthesized in laboratory	97	Isotactic polypropylene	4	[71]
4	Linear polystyrene Comb-like branched polystyrene	Synthesized	70.4 55.1	Isotactic polypropylene, grade T30S	1	[72]
5	Comb-like branched polystyrene Linear polystyrene Star shaped polystyrene	Synthesized	49.52 21.17 7.45	Isotactic polypropylene, grade T30S	1	[73]
6	Liquid crystal polyester (PBDPS)	Synthesized in laboratory	96.6	Isotactic polypropylene, grade T30S	4	[74]
7	Novolac	Qinan adhesive materials factory (China)	20.8	Isotactic polypropylene, grade F401	30	[75]
8	Polystyrene (PS) Styrene acrylonitrile copolymer (SAN)	PS from Taita Chemical Corp San from Mitsubishi plastics, Inc., Japan	26 32	Isotactic polypropylene, grade S1003	2	[76]
9	Liquid crystal polymer (LCP), Vectra A950	Hoechst IBERICA SA	23	Isotactic polypropylene	1	[77]
10	LCP-NA2	Synthesized in laboratory	70	Isotactic polypropylene powder TPH-XB-075	1	[78]

## Data Availability

The data presented in this study are available on request from the corresponding author.

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
