# Peer review of "β-Nucleated Polypropylene: Preparation, Nucleating Efficiency, Composite, and Future Prospects"

_polymers, 2023, doi:10.3390/polym15143107_

Round 1

Reviewer 1 Report

This Article devoted to the analysis of the features of the crystalline beta-form of polypropylene, the influence of its presence on the physicochemical properties of materials and composites, the review of the methods used to obtain the beta-form, the nucleating agents used and their efficiency increase. Given the significant effect of the content of this phase on the performance properties of materials and composites made of polypropylene, the tasks and goals set in the review are very relevant, important and of interest not only to specialists, but also to dilletants in this field of science. And, in this regard, it is doubly annoying that, unfortunately, the review did not work out in full.

When reading the Article, there is an ambivalent feeling. On the one hand, the Authors have carried out extensive research work, and certain results have been obtained. The above list of references is quite extensive and covers a wide time range. Despite the fact that the writing style is a little heavy and the Article is not easy to read, the Authors arranged the research material and discussions in a logical manner. But, on the other hand, many of these sections are very concise, often contain scattered facts, fragments of research (sometimes without references) and comments that are not united by any general conclusion or conclusion within this section. There is a review, but the style of presentation (long confusing sentences, use of scientific and non-scientific slang, excessive use of abbreviations without explanation, mention of various terms, effects and processes without reference and without explaining their essence, grammatical errors and omissions) make it difficult to read the article or do this process is uncomfortable.

Moreover, some sections are simply bewildering, such as the section on ethylene chains (line 243). The Authors do not at all explain the need for a critical analysis of such systems - why exactly ethylene, and not other fragments, and - most importantly - there is no conclusion on the totality of the analyzed material.

Further, making an attempt to systematize the currently used organic and inorganic nucleating agents (Table 1) and polymer analogs (Table 2), the Authors do not conduct any critical analysis or comparison with each other. Well, at least they could made the choice of the Authors for these systems. And for many nucleating agents, their abbreviation is not even fully deciphered, for many there is no efficiency data, which in principle casts doubt on the advisability of their inclusion in the tables.

And, in general, the Conclusion given by the Authors at the end of the article is of an exclusively general nature; it does not provide any specific conclusions or recommendations.

I believe that this Article requires serious revision, processing and reduction, and also correction of available typing errors before it would be suitable for publication in the Journal. In addition, the following grammatical and stylistic errors should be taken into account:

  - Throughout the text of the Article, the reference numbers and the word before it are written together, although this is not practiced in the Polymers journal.

- It should be remembered that the potential readers of this Article may be unfamiliar with the abbreviation of DCP (line 284), CRPPR (line 285),  abbreviation of “different nucleating agents” on Figure 3 and others.

- Line 36, no space between sentences.

- Reference in the text to Figure 1 (line 127) is much later than the figure shown. As a rule, all the Figures are positioned before their discussion and references on it. This is much harder to read the article.

- Line 121 and so on repeatedly in the text: beta must be subscript.

- Line 122, no space between words.

- there is no reference to figure 2 in the text.

- There are not enough notations in Figure 2, it is not clear what “nucleating agents” are.

- In figure 3, the name of the components is poorly visible and there is no explanation for them in the signature.

- Line 209. Slang – “People began…”

  - Confusing voluminous sentences "about nothing", lines 214-218:

“Different nucleating agent or complex in different kinds of polypropylene with different amount of different actual induced β-nucleation efficiency is not the same whether it is mentioned in the previous small molecule β-nucleating agent or compound or polymer β-nucleating agent. “

- Line 227-228 “The irregularity of the chain segments in the stereoscopic and regional effects…

  The authors refer to previous reviews on the crystallization of beta-polypropylene, but the purpose of this study becomes unclear. If this is a listing of beta-nucleating agents, then the Authors themselves refer to the previous reviews. In this case, I believe that the purpose of this work was a critical review of information, but this criticism, generalization, conclusions about the appropriateness of a particular approach in this review is not enough in my opinion. The work would noticeably benefit in terms of quality if it contained recommendations for readers.

  - Line 293 Text (and entire section) about nothing: "Another problem affecting the nucleation effect of β-crystal is the coexistence of polycrystalline polypropylene in addition to the great influence of the ethylene chain segment." No generalizing conclusions, just fragmentary materials with links, banal facts that an increase in the content of two polymorphs will lead to a decrease in the content of the third with a constant overall degree of crystallinity. There are quite decent and complete reviews on this topic, and the current one is only puzzling.

- Line 311 and onwards. The authors point out that "...The interaction between the nucleating agent and the PP matrix interface leads to the formation of β-crystals." One gets the impression that other polymorphs are formed somehow differently, not on the surface. This is true?

- Lines 394-395 The somewhat absurd sentence “We found that the nucleation efficiency of β-crystals is affected by many factors according to the summary of the previous studies.”

- - Lines 427-428. Incorrect offer. “Higher degree, wider distribution, and stronger nucleation ability[131,132]”.

- - Lines 441-442. Incomprehensible sentence: “At the same time, the grain is more refined, which is also suitable for the system of adding impact polypropylene, which can significantly improve the low temperature impact without reducing the rigidity.”

The English language of the Article also requires, in my opinion, adjustments. The Article is written by pretentious language, in a hard-to-read style and does not allow to understand the main points of the Article. The Author's desire to use long complex sentences leads to serious confusion. Although English is not my native language, I think that the Authors should paid attention to improve the English.

Author Response

Reviewer 1 This Article devoted to the analysis of the features of the crystalline beta-form of polypropylene, the influence of its presence on the physicochemical properties of materials and composites, the review of the methods used to obtain the beta-form, the nucleating agents used and their efficiency increase. Given the significant effect of the content of this phase on the performance properties of materials and composites made of polypropylene, the tasks and goals set in the review are very relevant, important and of interest not only to specialists, but also to dilletants in this field of science. And, in this regard, it is doubly annoying that, unfortunately, the review did not work out in full. Response: We thank this reviewer for the positive comments. When reading the Article, there is an ambivalent feeling. On the one hand, the Authors have carried out extensive research work, and certain results have been obtained. The above list of references is quite extensive and covers a wide time range. Despite the fact that the writing style is a little heavy and the Article is not easy to read, the Authors arranged the research material and discussions in a logical manner. But, on the other hand, many of these sections are very concise, often contain scattered facts, fragments of research (sometimes without references) and comments that are not united by any general conclusion or conclusion within this section. There is a review, but the style of presentation (long confusing sentences, use of scientific and non-scientific slang, excessive use of abbreviations without explanation, mention of various terms, effects and processes without reference and without explaining their essence, grammatical errors and omissions) make it difficult to read the article or do this process is uncomfortable. Response: Thank you for your comments. We have thoroughly reviewed the article according to your advices, and examined the terms in order to add readability to the layperson. Some abbreviations were also explained when they first appeared. We have added some general conclusions after some sections. Seen in after the section “Preparation Strategies of β-PP”, “Nucleating efficiency of β-crystal”, “Composite of PP with β nucleating agents”. Moreover, some sections are simply bewildering, such as the section on ethylene chains (line 243). The Authors do not at all explain the need for a critical analysis of such systems - why exactly ethylene, and not other fragments, and - most importantly - there is no conclusion on the totality of the analyzed material. Response: First, some unnecessary content has been deleted. The study of β nucleation of copolymerized polypropylene and random copolymerized polypropylene was cited, indicating that the ethylene segment has an effect on β nucleation. In addition, the peroxide preferentially attacked the tertiary carbon atoms close to the ethylene comonomer to improve the stereoregularity of the random copolymerized polypropylene. The above facts indicate that the ethylene chain segment plays an important role in the β nucleation effect. Finally, a general conclusion is drawn. Further, making an attempt to systematize the currently used organic and inorganic nucleating agents (Table 1) and polymer analogs (Table 2), the Authors do not conduct any critical analysis or comparison with each other. Well, at least they could made the choice of the Authors for these systems. And for many nucleating agents, their abbreviation is not even fully deciphered, for many there is no efficiency data, which in principle casts doubt on the advisability of their inclusion in the tables. Response: The results of Table 1 and Table 2 are compared, and the recommended choices are given. At the same time, the general conclusions are given. In addition, some abbreviations have been revised to make it easier for readers to understand. And, in general, the Conclusion given by the Authors at the end of the article is of an exclusively general nature; it does not provide any specific conclusions or recommendations. Response: The conclusion given by the authors at the end of the article has been revised. I believe that this Article requires serious revision, processing and reduction, and also correction of available typing errors before it would be suitable for publication in the Journal. In addition, the following grammatical and stylistic errors should be taken into account:   - Throughout the text of the Article, the reference numbers and the word before it are written together, although this is not practiced in the Polymers journal. Response: The reference numbers and the word before it have been separated from each other. - It should be remembered that the potential readers of this Article may be unfamiliar with the abbreviation of DCP (line 284), CRPPR (line 285), abbreviation of “different nucleating agents” on Figure 3 and others. Response: DCP (Line 284 in original manuscript, Line 323 in revised manuscript) and CRPPR (Line 285 in original manuscript, Line 325 in revised manuscript) have been spelled completely to make readers understand them easier. - Line 36, no space between sentences. Response: -Line 36 (Line 36 in original manuscript, Line 38 in revised manuscript), space have been added. - Reference in the text to Figure 1 (line 127) is much later than the figure shown. As a rule, all the Figures are positioned before their discussion and references on it. This is much harder to read the article. Response: The position of Figure 1 (Line 127 in original manuscript, Line 123 in revised manuscript) has been adjusted after discussion. - Line 121 and so on repeatedly in the text: beta must be subscript. Response: β has been labeled as a subscript. - Line 122, no space between words. Response: -Line 122 (Line 122 in original manuscript, Line 113 in revised manuscript), space have been added. - there is no reference to figure 2 in the text. Response: Figure 2 in the text has been mentioned in line 182 (in revised manuscript) and the corresponding reference is 44. - There are not enough notations in Figure 2, it is not clear what “nucleating agents” are. Response: Figure 2 is quoted from reference 44 and can’t been edited. The nucleating agent is the monetaria moneta powder described in Figure 2 caption. - In figure 3, the name of the components is poorly visible and there is no explanation for them in the signature. Response: Figure 3 is quoted from reference 41. The image has been replaced by the high-res image which was been downloaded from https://ars.els-cdn.com/content/image/1-s2.0-S0014305717301702-gr2_lrg.jpg. An explanation has been added in the text. - Line 209. Slang – “People began…”   - Confusing voluminous sentences "about nothing", lines 214-218: “Different nucleating agent or complex in different kinds of polypropylene with different amount of different actual induced β-nucleation efficiency is not the same whether it is mentioned in the previous small molecule β-nucleating agent or compound or polymer β-nucleating agent. “ Response: - Line 209. (in original manuscript) Slang – “People began…” has been revised as “Polymer β-nucleating agents are gradually becoming a research hotspot in recent years.” in lines 241-242 (in revised manuscript). Lines 214-218 (in original manuscript) have been revised as “Different nucleating agents have different β-nucleation efficiency in polypropylene with the same addition amounts. On the other hand, a nucleating agent has different β-nucleation efficiency in different kinds of polypropylene with the same addition amounts.” in lines 267-270 (in revised manuscript). - Line 227-228 “The irregularity of the chain segments in the stereoscopic and regional effects…   The authors refer to previous reviews on the crystallization of beta-polypropylene, but the purpose of this study becomes unclear. If this is a listing of beta-nucleating agents, then the Authors themselves refer to the previous reviews. In this case, I believe that the purpose of this work was a critical review of information, but this criticism, generalization, conclusions about the appropriateness of a particular approach in this review is not enough in my opinion. The work would noticeably benefit in terms of quality if it contained recommendations for readers. Response: For this section, the structure of the text has been changed. First, the facts of the literature are cited, and finally a conclusion has been drawn. “The chain structure changes the number of crystal centers and affects the spherulite size. The irregularity of the chain segments in the stereoscopic and regional effects, as well as the insertion of comonomers have a negative effect on the β-crystal efficiency in the polymer matrix.” Seen in Lines 291-294 (in revised manuscript).   - Line 293 Text (and entire section) about nothing: "Another problem affecting the nucleation effect of β-crystal is the coexistence of polycrystalline polypropylene in addition to the great influence of the ethylene chain segment." No generalizing conclusions, just fragmentary materials with links, banal facts that an increase in the content of two polymorphs will lead to a decrease in the content of the third with a constant overall degree of crystallinity. There are quite decent and complete reviews on this topic, and the current one is only puzzling. Response: For this section, the structure of the text has been changed. By citing the literature, we state the facts of polycrystals. Based on the facts, we conclude “the influence factors of α or γ crystal should be minimized. Increasing the nucleation site of β-crystals or appropriate processing conditions can make the induction effect of β-crystal dominant. This helps to obtain polypropylenes with a high β-crystal content, despite polycrystalline also existing.” Seen in Lines 354-357 (in revised manuscript). This is also the direction of researchers' efforts to obtain high β-crystals. - Line 311 and onwards. The authors point out that "...The interaction between the nucleating agent and the PP matrix interface leads to the formation of β-crystals." One gets the impression that other polymorphs are formed somehow differently, not on the surface. This is true? Response: We’re sorry this is an expression problem in English. We have deleted the words that are easy to misunderstand “The interaction between the nucleating agent and the PP matrix interface leads to the formation of β-crystals”. At the same time, some revisions were made. - Lines 394-395 The somewhat absurd sentence “We found that the nucleation efficiency of β-crystals is affected by many factors according to the summary of the previous studies.” Response: - Lines 394-395 (Line 441 in revised manuscript) have been revised as “The nucleation efficiency of β-crystals is affected by many factors.” - - Lines 427-428. Incorrect offer. “Higher degree, wider distribution, and stronger nucleation ability[131,132]”. Response: We’re sorry this is an expression problem in English. Lines 427-428 (Lines 475-477 in revised manuscript) have been revised as “When the molecular weight is above a certain degree, the higher isotacticity and the wider molecular weight distribution, the stronger the β-crystallization ability”. - - Lines 441-442. Incomprehensible sentence: “At the same time, the grain is more refined, which is also suitable for the system of adding impact polypropylene, which can significantly improve the low temperature impact without reducing the rigidity.” Response: We have the deleted the Incomprehensible sentence. The summary at the end has well expressed the effect of adding homopolymer. The English language of the Article also requires, in my opinion, adjustments. The Article is written by pretentious language, in a hard-to-read style and does not allow to understand the main points of the Article. The Author's desire to use long complex sentences leads to serious confusion. Although English is not my native language, I think that the Authors should paid attention to improve the English. Response: The article has been undergone extensive English revisions in https://www.mdpi.com/authors/english. The English editing ID is english-68436 and the certificate has been downloaded. Thank you for the professional and valuable comments, we have made changes according to your comments, the changes have been highlighted, you can see whether the changes are satisfactory.

Reviewer 2 Report

Comments

The comments for the work presented in the manuscript entitled, ‘Β-Nucleated Polypropylene: Preparation, Nucleating Efficiency, Composite and Future Prospects’, are given below:

1.      Figure 1a, XRD spectra, it is mentioned that the planes corresponding to (110), (040) and (130) denotes a significant change. But the spectra do not show any peaks in this region. Explanation is needed for this. Moreover, the XRD spectra was taken as the reference from this paper (https://doi.org/10.1021/acs.macromol.1c01038), where the peaks are clearly visible and there is no citation of this reference.

2.      In Figure 1b, DSC exotherms were observed at two different temperatures, i.e., 150 and 165 C. All these exotherms looks similar and there are no differences observed in the area within the peak. Calculation of enthalpy (∆H) is needed to get an idea about the amount of heat liberated. This could be done straightaway from the instrument. Enthalpy values for all the peaks at these two temperatures should be reported.

3.      Figure 3 shows the SEM images of four different samples but at different magnifications. It is very hard to observe the changes in different magnifications. Re-insert SEM images of these samples at same magnifications.

4.      The language throughout the manuscript should be checked for typo and grammatical errors, and re-write the sentences wherever required.

5.      The manuscript shows 47% plagiarism with this paper (https://doi.org/10.1021/acs.macromol.1c01038). Modify the manuscript accordingly to minimize the plagiarism.

After these modifications, the manuscript can be accepted for publication.

Comments

The comments for the work presented in the manuscript entitled, ‘Β-Nucleated Polypropylene: Preparation, Nucleating Efficiency, Composite and Future Prospects’, are given below:

1.      Figure 1a, XRD spectra, it is mentioned that the planes corresponding to (110), (040) and (130) denotes a significant change. But the spectra do not show any peaks in this region. Explanation is needed for this. Moreover, the XRD spectra was taken as the reference from this paper (https://doi.org/10.1021/acs.macromol.1c01038), where the peaks are clearly visible and there is no citation of this reference.

2.      In Figure 1b, DSC exotherms were observed at two different temperatures, i.e., 150 and 165 C. All these exotherms looks similar and there are no differences observed in the area within the peak. Calculation of enthalpy (∆H) is needed to get an idea about the amount of heat liberated. This could be done straightaway from the instrument. Enthalpy values for all the peaks at these two temperatures should be reported.

3.      Figure 3 shows the SEM images of four different samples but at different magnifications. It is very hard to observe the changes in different magnifications. Re-insert SEM images of these samples at same magnifications.

4.      The language throughout the manuscript should be checked for typo and grammatical errors, and re-write the sentences wherever required.

5.      The manuscript shows 47% plagiarism with this paper (https://doi.org/10.1021/acs.macromol.1c01038). Modify the manuscript accordingly to minimize the plagiarism.

After these modifications, the manuscript can be accepted for publication.

Author Response

Reviewer 2

The comments for the work presented in the manuscript entitled, ‘Β-Nucleated Polypropylene: Preparation, Nucleating Efficiency, Composite and Future Prospects’, are given below:

  1. Figure 1a, XRD spectra, it is mentioned that the planes corresponding to (110), (040) and (130) denotes a significant change. But the spectra do not show any peaks in this region. Explanation is needed for this. Moreover, the XRD spectra was taken as the reference from this paper (https://doi.org/10.1021/acs.macromol.1c01038), where the peaks are clearly visible and there is no citation of this reference.

Response: This paper (https://doi.org/10.1021/acs.macromol.1c01038) is an important reference that we have cited as reference 55. Fig. 1a is quoted from reference 41. It discloses that all the four samples exhibited two characteristic diffraction peaks at 2θ =16.1° and 21.2° which correspond to the (300) and (301) crystal plane of β-form, and the diffraction peaks at 14.1°, 16.9°, and 18.6° which correspond to the α (110), α (040), and α (130) were undetectable, indicating that all the four precursor films were composed of almost pure β-crystals. Seen in lines 119-123 (in revised manuscript).

  1. In Figure 1b, DSC exotherms were observed at two different temperatures, i.e., 150 and 165 C. All these exotherms looks similar and there are no differences observed in the area within the peak. Calculation of enthalpy (∆H) is needed to get an idea about the amount of heat liberated. This could be done straightaway from the instrument. Enthalpy values for all the peaks at these two temperatures should be reported.

Response: Figure 1b is quoted from reference 41. The author of the reference did not identify the enthalpy of the two peaks, so we can’t edit it. The purpose of quoting the figure is to illustrate that the calculation of β crystal content can be carried out by DSC curve.

  1. Figure 3 shows the SEM images of four different samples but at different magnifications. It is very hard to observe the changes in different magnifications. Re-insert SEM images of these samples at same magnifications.

Response: Figure 3 is quoted from reference 41. The high-res image has been downloaded from https://ars.els-cdn.com/content/image/1-s2.0-S0014305717301702-gr2_lrg.jpg. We can’t edit the magnification. Despite different magnifications, different morphologies of β-crystal induced by different nucleating agents also can be observed. In order to better explain the morphology, we have supplemented a text description for a better understanding in lines 199-204 (in revised manuscript).

  1. The language throughout the manuscript should be checked for typo and grammatical errors, and re-write the sentences wherever required.

Response: The article has been undergone extensive English revisions in https://www.mdpi.com/authors/english. The English editing ID is english-68436 and the certificate has been downloaded.

  1. The manuscript shows 47% plagiarism with this paper (https://doi.org/10.1021/acs.macromol.1c01038). Modify the manuscript accordingly to minimize the plagiarism.

Response: This paper (https://doi.org/10.1021/acs.macromol.1c01038) is an important reference that we have cited as reference 55. The research of this article is very fruitful and helpful to us. The DCHT they studied has a very high β conversion rate, so it is also recommended to use in our article in line 162 (in revised manuscript). We also checked and compared the structure of our article and the reference paper (https://doi.org/10.1021/acs.macromol.1c01038), made some modifications and felt that there was not much repetition. Thank you again for your advice.

After these modifications, the manuscript can be accepted for publication.

Thank you for the professional and valuable comments, we have made changes according to your comments, the changes have been highlighted, you can see whether the changes are satisfactory.

Round 2

Reviewer 1 Report

In general, I am satisfied with the correction of the Article made by the Authors. The Article looks more significant and contains certain discussions and conclusions. I believe that the Article can be published in the Polymers journal in its current form. The only remark concerns the section of Article 3.1.3 devoted to the effect of polymorphism on crystallization.

The Authors somewhat confuse the concepts of polycrystallinity and polymorphism. If polymorphism implies the existence of several crystalline forms (namely, crystalline forms) of a compound, then polycrystallinity simply means the presence in the sample of not one single crystal, but many much smaller crystallites in a certain size range. This actually happens in polymeric crystallizing samples and in polypropylene samples as well (obtaining monocrystals of polymers is a separate story).

Therefore, the phrases on lines 341-343 sound somewhat incorrect: “Another problem affecting the nucleation effect of β-crystals is the coexistence of polycrystalline polypropylene in addition to the great influence of the ethylene chain segment…”, as well as on lines 345-347 "However, the coexistence of polycrystals will still affect the main crystal form, just as it is difficult to obtain a 100% β-crystal PP [43,82]" and lines 356-357 "This helps to obtain polypropylenes with a high β-crystal content, despite polycrystalline also existing.”

I would like to suggest the Authors to correct the sentences as follows:

Lines 341-343: "Another problem affecting the nucleation effect of β-crystals is the coexistence of different polymorphic forms of polypropylene in addition to the great influence of the ethylene chain segment."

Lines 345-347: "However, the coexistence of different polymorphic forms will still affect the main crystal form, just as it is difficult to obtain a 100% β-crystal PP [43,82]."

Lines 356-357: "This helps to obtain polypropylenes with a high β-crystal content, despite other polymorphs of PP also existing."

Reviewer 2 Report

The author has addressed in full all the concerns of reviewers. I recommend that the revised manuscript could be published in polymers

 Minor editing of English language required